# Does educational level predict hearing aid self-efficacy in experienced older adult hearing aid users from Latin America? Validation process of the Spanish version of the MARS-HA questionnaire

Eduardo Fuentes-López[1]*, Adrian Fuente[2,3], Gonzalo Valdivia[4], Manuel Luna-Monsalve[5]

**1** Carrera de Fonoaudiología, Departamento de Ciencias de la Salud, Facultad de Medicina, Pontificia Universidad Católica de Chile, Santiago, Chile, **2** École d'orthophonie et d'audiologie, Faculté de Médecine, Université de Montréal, Montréal, Québec, Canada, **3** Centre de Recherche de l'Institut Universitaire de Gériatrie de Montréal, Montréal, Québec, Canada, **4** Departamento de Salud Pública, Facultad de Medicina, Pontificia Universidad Católica de Chile, Santiago, Chile, **5** Escuela de Fonoaudiología, Facultad de Ciencias de la Salud, Universidad San Sebastián, Concepción, Chile

* eduardo.fuentes@uc.cl

## Abstract

Hearing aids are the most common rehabilitation strategy for age-related hearing loss. However, 25% to 50% of older adults fitted with hearing aids do not wear them post-fitting. Hearing aid self-efficacy has been suggested as one of the key factors that may explain adherence to hearing aids in older adults. The primary aim of this study was to determine a possible association between educational level and hearing aid self-efficacy in older adult hearing aid users from a Latin American country (i.e., Chile). The secondary aim was to determine if in this sample of older adults, hearing aid self-efficacy predicted hearing aid adherence as previously suggested by other studies. The MARS-HA (Measure of Audiologic Rehabilitation Self-Efficacy for Hearing Aids) questionnaire was used to measure hearing aid self-efficacy. This questionnaire was initially adapted into Spanish (S-MARS-HA) using forward and backward translations by bilingual English-Spanish speakers. A sample of 252 older adults fitted with hearing aids at a public hospital in Santiago, Chile, was investigated. Educational level was measured as the number of years of formal education. Participants responded to the S-MARS-HA along with questions exploring social support, attitudes in using hearing aids, participation in social events, and vision and joint problems. Hearing aid adherence was investigated with the use of a question from the International Outcome Inventory for Hearing Aids. All these procedures were conducted at the participants' homes. Pure-tone average (PTA; 500–4000 Hz) in the fitted ear was obtained from the participants' medical records. Univariate and multivariate regression models were constructed to investigate the association between educational level and hearing aid self-efficacy controlling for the covariates of interest (e.g., social support, attitudes in using hearing aids, PTA). The S-MARS-HA showed an adequate construct validity along with a good reliability. Results of the multivariate regression analyses showed that educational level significantly predicted

**Data Availability Statement:** All relevant data are within the manuscript and its Supporting Information files.

**Funding:** This work was supported by: EFL; 1. Concurso Especial de Investigación Semilla-Interdisciplinario in 2015 from Pontificia Universidad Católica de Chile, Project grant: PS 05/15, https://medicina.uc.cl/investigacion/didemuc-informa/ano-2015/didemuc-no-6-concurso-de-investigacion-semilla-interdisciplinario/; 2. National Commission for Scientific and Technological Research (CONICYT) under the National Fund for Research and Development in Health (Fondo Nacional de Investigación y Desarrollo en Salud, FONIS) Project grant: SA16I0290, https://www.conicyt.cl/fondef/lineas-de-programa/instrumentos-vigentes/concurso-nacional-de-proyectos-de-investigacion-y-desarrollo-en-salud-fonis/. The funders had no role in study design, data collection and analysis, decision to publish, or preparation of the manuscript.

**Competing interests:** The authors have declared that no competing interests exist.

hearing aid self-efficacy. Covariates significantly associated with this outcome included attitudes in using hearing aids and PTA in the fitted ear. Finally, a significant association between hearing aid self-efficacy and adherence to hearing aid use was observed. In conclusion, this study showed a significant association between educational level and hearing aid self-efficacy in older adults from a developing Latin American country. Thus, this variable should be considered when designing and delivering aural rehabilitation programs such as hearing aids to older adults, especially those from developing countries.

## Introduction

Hearing aids are the most common rehabilitation strategy for age-related hearing loss. However, 25% to 50% of older adults fitted with hearing aids do not wear them post-fitting [1–3]. A number of studies have been conducted to determine the variables that may explain why older adults stop using hearing aids [4–6]. Hearing aid self-efficacy has been suggested as one of the key factors that may explain adherence to hearing aids in older adults [5, 7–9].

Hearing aid self-efficacy has been defined by Smith and West [10] as the individuals' confidence about their own ability to look after and use a hearing aid successfully. Meyer et al. [9] suggested that the greater the degree of hearing aid self-efficacy is, the longer the patient will continue to use the device. Hickson et al. [8] concluded that participants with more positive attitudes to hearing aids and higher levels of self-efficacy were more likely to be successful hearing aid users.

Therefore, understanding the variables that can predict hearing aid self-efficacy in older adults seems to be crucial for successful hearing aid fitting in the clinical setting. Smith and West [11] found that individuals with moderately severe hearing loss had lower self-efficacy than individuals with mild hearing loss. Individuals with poor unaided word recognition abilities in quiet also had lower overall hearing aid self-efficacy than individuals with good to fair unaided word recognition abilities. Meyer et al. [9] found that a positive experience with the hearing aid, along with having no visual disability and having positive support from a significant other, was associated with adequate hearing aid self-efficacy in older adult hearing aid owners.

It should be noted, however, that the studies mentioned above have been conducted in developed countries where people's educational levels are higher than developing countries [12]. In addition, in regions such as Latin America, educational level is unequally distributed [13]. Thus, we hypothesize that in the context of developing countries, educational level may be a predictor for hearing aid self-efficacy and, ultimately, for hearing aid adherence. This is because the readability of the written materials, such as hearing aid user manuals, is likely to be associated with educational level [14]. Hearing aid user manuals provide indications directly related to aspects associated with hearing aid self-efficacy such as basic and advanced handling of the device. More specifically, the effect of educational level on self-efficacy is related to how individuals formulate self-efficacy beliefs. According to Bandura [15–18], individuals formulate self-efficacy beliefs primarily from four sources of information: (1) enactive mastery experiences (judging one's own capabilities for a desired behavior by performing the skills necessary to achieve that behavior), (2) vicarious experience (based on their observations of the experiences of others), (3) verbal persuasion (forms of persuasion, both verbal and nonverbal), and (4) physiological and affective states (observing their own physiological and emotional reactions). It is possible that the effect of educational level on self-efficacy mainly occurs

through *"enactive mastery experiences"* and *"verbal persuasion"* sources of information. One strategy to increase the *"Enactive Mastery Experience"* is to divide new skills into smaller steps [11] as it is incorporated in the hearing aid user guides for older people, whereas "verbal persuasion" can be increased by giving appropriate feedback or providing pedagogic materials [11], such as the ones provided in follow-up sessions. In both cases, having a higher educational level would facilitate the aforementioned strategies to increase both "mastery experience" and "verbal persuasion" sources of information.

Previous studies conducted in developed countries have not investigated the possible association between educational level and hearing aid self-efficacy in older adults [9, 19]. However, the educational level showed to be associated with hearing aid adherence in developed countries [20–22]. Such an association may depend on the strong relationship between educational level and income levels [20]. People who have achieved higher levels of education are likely to have higher annual income levels and thus they can afford paying for the hearing aids and other related components such as the batteries [23].

Therefore, the primary aim of this study was to determine a possible association between educational level and hearing aid self-efficacy in older adult hearing aid users from a Latin American country (i.e., Chile), controlling for the covariates previously suggested to predict hearing aid self-efficacy in this patient population. Such covariates include hearing thresholds, social support, participation in social events since getting a hearing aid, age, and vision and joint problems. The secondary aim was to determine if in this sample of older adults, hearing aid self-efficacy predicted hearing aid adherence as observed in previous studies conducted in developed countries (e.g., Hickson et al.,[8]; Meyer et al., [9]; Ng & Loke, [5]).

Because no previously validated instruments in Spanish were available to evaluate hearing aid self-efficacy, the MARS-HA questionnaire [10] was initially adapted to be used with Spanish-speaking older adults. This questionnaire includes subscales that allow for self-efficacy to be evaluated in a number of aspects such as basic and advanced handling. In its original English version, the MARS-HA questionnaire has adequate construct validity [10]. In addition, it has been used in a number of studies investigating hearing aid self-efficacy in English-speaking populations (e.g., Convery et al., [24]; Hickson et al., [8]; Ferguson et al., [25]; Kelly-Campbell & McMillan,[19]; Vincent et al., [26]).

## Materials and methods

The protocol for this study was approved by the Ethics Committee of La Florida Metropolitan Hospital Clinic (Chile) and the Ethics Committee of the Pontificia Universidad Católica de Chile. All participants signed an informed consent form.

### Initial step: Cross-cultural adaptation of the Measure of Audiologic Rehabilitation Self-Efficacy for Hearing Aids (MARS-HA) questionnaire

First, two bilingual English-Spanish speakers (native Spanish speakers) independently translated the English version of the MARS-HA questionnaire into Spanish. Then two audiologists merged both translations to produce a single document in Spanish. The differences in the translations were owing mainly to the terminology and, to a lesser extent, grammar. This document was then translated back into English by a bilingual Spanish-English speaker (a native English speaker). Next, a panel of experts composed of a professional translator, a linguist, and an audiologist reviewed both the forward and backward translations, as well as the original version of the MARS-HA in English. This expert committee elaborated a preliminary version of the Spanish MARS-HA (S-MARS-HA), taking into consideration the semantic, idiomatic, and conceptual equivalence between the original English version and the newly created

Spanish version of the questionnaire. Certain technical words were replaced by lay terms in Spanish (e.g., "dispositivo auditivo" was replaced with "audifono" [hearing aid]). In addition, grammatical structures were modified so that conditional phrases all began in the same way (for example, "Si usara audífonos . . ." [If I wore hearing aids . . .]). Also, the structure of the responses was simplified to short phrases, like how "moderadamente seguro" (moderately sure) was replaced with "podría hacerlo" (I could do it). A graphic representation of these response options was constructed similar to that used in the original version. After these modifications were made, a preliminary version of the S-MARS-HA was obtained.

A pilot study involving 48 older adults was carried out to test the preliminary version of the S-MARS-HA, in which the participants were asked if the questions were clear. In addition, the participants' responses were reviewed to determine if there were any questions left blank or if the same answer was used repeatedly for each question. The participants did not mention any unclear or confusing questions, and no instances of blank or repeated answers were found. Therefore, no further modifications were made to the S-MARS-HA (Annex 1).

Like the original version in English, the S-MARS-HA was composed of 24 items grouped into four different dimensions/factors: basic handling (items 1, 2, 3, 4, 5, 7, and 10), advanced handling (items 6, 8, 9, 11, and 12), adjustment (items 13, 14, and 15), and aided listening (items 16, 17, 18, 19, 20, 21, 22, 23, and 24). The questionnaire had a continuous reply scale, starting at 0% when the person indicated that they could not carry out a certain action at all, to 50% for those who could do the indicated task partially, and reaching 100% when they were certain they could do the indicated task (S1 Annex). As previously mentioned, a graphic representation of the possible answers (from 0% to 100%) was included, as in the instrument's original version.

## Main study: The effect of educational level on hearing aid self-efficacy in a group of older adults from Chile

**Sample.**   The sample comprised 252 older adults who had been fitted with hearing aids at least one year prior to the beginning of the study. Thus, participants had enough experience with the device when answering the S-MARS-HA. All participants were between 65 and 85 years of age and were fitted with one hearing aid at La Florida Metropolitan Hospital Clinic in Santiago, Chile. This is a public hospital, so patients pay up to 20% of the cost of the hearing aid.

The sample size was calculated for the validity study of the S-MARS-HA. The validity of the S-MARS-HA was obtained using confirmatory factor analysis. Thus, to calculate the sample size, the method based on the confirmatory factor analysis's goodness of fit suggested by Mac-Callum, Browne and Sugawara was used [27]. A root mean square error of approximation (RMSEA) test was used to obtain the sample size needed to test the close-fit hypothesis of the confirmatory factorial model. In this approach, the RMSEA test value should be set at a level that reflects a close-fit model (i.e., RMSEA < 0.05). Thus, considering an RMSEA = 0.04 with a 90% power and $\alpha$ = 0.05, the sample size required was 220 participants. Given the chance of possible dropouts (15% of the sample), a final sample of 252 participants was sought. This sample size was considered to be sufficient enough to run multivariate linear regression models with nine potential predictors obtaining 90% power with an $\alpha$ = 0.05 for the estimation of an effect size $f^2$ = 0.04 (small to moderate).

**Sample selection procedures.**   La Florida Metropolitan Hospital Clinic began a hearing aid program for older adults in 2015. This is a program funded by the Chilean Ministry of Health, and older adults who require hearing aids are monaurally fitted with one device [28]. The criterion for providing a hearing aid under this program is the diagnosis of a bilateral

moderate hearing loss (i.e., pure-tone average 500–4000 Hz equal or worse than 40 dB HL in the better ear). At the time of this study, 823 older adults had been fitted with a hearing aid for the first time at this hospital. From this list, those who had been fitted with a hearing aid at least a year ago were randomly chosen and then contacted by telephone to invite them to participate in the study. Those who agreed to participate and who authorized the review of their clinical records were then preselected. Their clinical files were accessed and reviewed to see whether they presented with hearing problems other than age-related hearing loss (e.g., onset of hearing loss before age 60 or history of otitis media). Those with no such a history were then contacted by telephone to schedule a visit to their homes.

**Home interview and evaluation.**   Older adults who accepted to be interviewed and evaluated at their homes were visited by one of the audiologists previously trained by the research team. This home visit lasted for about one and a half hours. Initially, a shortened version of the mini-mental state examination (MMSE) that was previously adapted and validated in Chile was applied [29]. The aforementioned instrument has been used in studies conducted in both Chile [30] and other Latin American countries [31–34]. The maximum overall score for this screening tool is 19. A cognitive impairment is suspected when a person obtains a score equal to or less than 12 points. Therefore, participants with a score equal to or below 12 were excluded from the study. In addition, participants with communicative difficulties unrelated to hearing problems (e.g., aphasia) were also excluded. Selected participants continued with the instruments described below.

**S-MARS-HA.**   The questionnaire, containing seven pages, and a pencil were handed to the participants, inviting them to complete it. Explanations about how to answer the questions were provided. A graphic representation of the S-MARS-HA response options, as well as those of the other instruments (see below), was in print form, with a font size large enough to be easily read (Arial font, size 40). On average, they completed the questionnaire in 20 minutes. In case an older adult reported poor eyesight (even when wearing eyeglasses), the questions and possible responses for the S-MARS-HA (and for the questions of the other instruments, see below) were read aloud by the interviewer. The participants could indicate their answers either verbally or by pointing to the graphical representation. The interviewers checked if any questions were left blank. If so, participants were asked to answer such questions.

**Educational level.**   Years of formal education were obtained with the use of two standardized questions, which are part of the Chilean national survey for older adults [34]. These questions were (1) what is the highest educational level you have reached and (2) how many years did you attend school, including tertiary studies. Some participants did not directly recall the number of years they attended school, and thus, based on question 1, the number of years of formal education was obtained. In case the participants did not complete a certain educational level (preparatory, high school, or tertiary studies), they were assigned a number of years according to the last grade they reached. Thus, both questions were used to cross-check the number of years of formal education the participants had. These questions were applied in another study about hearing aid use in Chile [30].

**Adherence to hearing aids.**   Adherence to hearing aid use was determined through one of the questions from the Spanish version of the International Outcome Inventory for Hearing Aids (IOI-HA) [35]: "Think about how much you used your present hearing aid(s) over the past two weeks. On an average day, how many hours did you use the hearing aid(s)?" In an ordinal scale, the possible answers went from none to less than an hour a day, 1 to 4 hours a day, 4 to 8 hours a day, and more than 8 hours a day. Answers were coded from 5 points (i.e., more than 8 hours a day) to 1 point (i.e., none).

**Other variables.**   Variables other than educational level were also explored to be included as covariates in the regression models. Age and the report of vision and joint problems were

obtained with the use of standardized questions which are part of the Chilean national survey for older adults [34]. In the case of vision problems, the question asked was "Wearing glasses or contact lenses, how would you describe your sight?" The five possible closed-format answers ranged from *"very good"* (5 points) to *"very bad"* (1 point). The possible presence of joint problems was explored with the following question: "Has a doctor ever told you that you have arthritis, osteoporosis, osteoarthritis, or joint problems?" The possible answers were "yes" or "no".

Hearing thresholds were obtained from the participants' clinical records. Audiologists at La Florida Hospital evaluated the participants' hearing threshold from 250 to 8000 Hz in a sound-proof booth according to ISO 8253–1 standard. For analysis purposes, the pure-tone average (PTA) for the aided ear was obtained considering hearing thresholds at 500, 1000, 2000, and 4000 Hz.

Changes in social support after getting the hearing aid were evaluated with three questions from the Glasgow Benefit Inventory (GBI) questionnaire [36]: (1) "Since getting your hearing aid, do you feel that you have had more or less support from your friends?" (The possible answers ranged from "much more support" with 5 points to "much less support" with 1 point); (2) "Since getting your hearing aid, do you feel that you have had more or less support from your family?" (The possible answers ranged from "much more support" with 5 points to "much less support" with 1 point); and (3) "Since getting your hearing aid, are there more or fewer people who really care about you?" (The possible answers range from "many more people" with 5 points to "many fewer people" with 1 point).

Changes in social participation after getting the hearing aid were evaluated with the following question from the GBI [36]: "Since getting your hearing aid, have you been able to participate in more or fewer social activities?" The possible answers ranged from "many more activities" with 5 points to "many fewer activities" with 1 point.

Attitudes in using hearing aids were evaluated through the question used by Hickson et al. [8] in a study with Australian older adults: "How would you evaluate your attitude to use a hearing aid?" The possible options were presented on a scale of -5 (very negative) to +5 (very positive).

## Statistical analysis

**Psychometric properties of the S-MARS-HA (validity and reliability).** The construct validity of the S-MARS-HA was obtained using a confirmatory factor analysis that specified a model in which the items were grouped into four factors identified by the authors of the original instrument. Diverse goodness-of-fit indices were obtained [37]: (a) the comparative fit index (CFI), related to the degree of correlation among the questionnaire items; (b) the Tucker-Lewis index (TLI), which penalizes complex models; and (c) the root mean square error of approximation (RMSEA), which quantifies the model's lack of adjustment. The hypothesis that the model had an adequate adjustment was evaluated using the close-fit test ($H_0$: RMSEA $\leq$ 0.05). In addition, the determination coefficient ($R^2$) was estimated, quantifying the variable's percentage of variance, explained by the latent factor [37]. Internal consistency was evaluated through correlations among the four factors and the overall score (Spearman's rho). Reliability was also estimated through the Cronbach's alpha coefficient.

**Association between educational level and hearing aid self-efficacy.** A univariate regression model was created between the number of years of formal education (continuous variable) and hearing aid self-efficacy. Additionally, univariate regression models were created for each of the covariates of interest: age (continuous variable), each of the three questions of the GBI about changes in social support after being fitted with the hearing aid (categorical

variable with five possible categories), joint problems (categorical variable, answerable by "yes" or "no"), vision problems (categorical variable with five possible categories), attitude in using hearing aids (10-point scale, from very negative [-5 points] to very positive [+5 points]), pure-tone average (PTA) for the aided ear (continuous variable), and changes in social participation after getting the hearing aid (categorical variable with 5 possible categories). Univariate regression models were created for the S-MARS-HA overall score and the score for each of the four subscales (i.e., basic handling, advanced handling, adjustment, and aided listening). Then multivariate regression models were constructed with the factors significantly associated with the dependent variable (overall S-MARS-HA score and scores for each of the four S-MARS-HA subscales) in the univariate models. Thus, five multivariate regression models were constructed.

For all univariate and multivariate models, because the response variable (i.e., scores for the overall and subscales of the S-MARS-HA) did not have a normal distribution, the standard error was estimated using bootstrapping. Thus, 95% confidence interval (CI) was obtained using the bias-corrected and accelerated method [38].

**Association between hearing aid self-efficacy and hearing aid adherence.** With the aim to investigate the possible effect of hearing aid self-efficacy on hearing aid adherence, univariate ordinal regression models were constructed. The models included adherence to hearing aids (IOI-HA scores, five ordinal categories) as the outcome measure and the independent variable of hearing aid self-efficacy (S-MARS-HA overall score and scores for each of the four S-MARS-HA subscales). The Mplus v.7.3 and STATA v.14 were used for all statistical analyses.

## Results

### Descriptive statistics

The average age of the participants (n = 252) was 74.5 years, and their average number of years of formal education was 8.6 (95% CI 8.1–9.2) (Table 1). Almost 50% mentioned they had poor eyesight, and 30% reported bad or very bad eyesight (S1 Dataset). All participants had either

**Table 1. General characteristics of the sample of older adult, hearing aid users (n = 252)[a].**

| | |
|---|---|
| Age (years) | 74.5 (73.7–75.2) |
| Percentage of women in the sample | 51.6 (45.4–57.7) |
| Number of years of formal education | 8.6 (8.1–9.2) |
| Percentage of participants who reported one or more of the following health conditions: arthritis, osteoporosis, osteoarthritis and joint problems | 50.0 (43.8–56.2) |
| Vision (self-reported) | |
| Very good | 2.4 (1.1–5.2) |
| Good | 20.2 (15.7–25.7) |
| Mediocre | 47.6 (41.5–53.8) |
| Bad | 23.4 (18.6–29.1) |
| Very bad | 6.4 (3.9–10.1) |
| Pure-tone average (PTA, 500–4000 Hz)) in the aided ear (dB HL) | 55.7 (53.9–57.6) |

[a] Values are expressed as average or relative frequency, as appropriate, with a 95% CI.

asymmetrical or symmetrical sensorineural hearing loss of different degrees. The right ear PTA (500, 1000, 2000, and 4000 Hz) was 59.2 dB HL, and the left ear PTA was 58.5 dB HL. No significant differences for PTA between both ears ($p = 0.48$) were observed.

## Psychometric properties of the S-MARS-HA

The overall high scores of the S-MARS-HA and its four subscales stand out. In every case, the mean was above 50%, with the highest scores in the *basic handling* and *adjustment* subscales, with the mean being above 90% in both subscales (Table 2). All the subscales were significantly correlated with the questionnaire's overall score (showing high internal consistency). The strongest correlations were obtained between the overall score and the scores of the *basic handling* and *advanced handling* subscales (*rho* = 0.74 and *rho* = 0.77, respectively). Reliability for the overall questionnaire was found to be high (Cronbach's alpha = 0.88). The Cronbach's alpha obtained for each of the S-MARS-HA subscales varied between 0.90 and 0.72 (Table 2).

## Construct validity

In the confirmatory factor analysis, the factorial model of the four factors (i.e., subscales) had adequate adjustment indices (Table 3). Both the CFI and the TLI indices were high, reaching 0.94. The RMSEA value (0.049) was within a good fit range ($< 0.05$) at the upper limit of its confidence interval (0.058) within the acceptable fit range (Wang & Wang, 2012). Considering that the close-fit test was not statistically significant (*p = 0.56*), it was not possible to reject the goodness-of-fit hypothesis. To improve the fit, the error variances were freely estimated for the association between items 1 and 2, items 4 and 5, and items 21 and 22. All the standardized factorial loads were greater than 0.40 and were statistically significant ($p < 0.001$). In the case of $R^2$, all the values were higher than 0.30, with the exception of items 11 and 16.

**Table 2. Descriptive statistics along with reliability coefficients for the overall and subscales of the Spanish version of the Measure of Audiologic Rehabilitation Self-efficacy for Hearing Aids questionnaire (S-MARS-HA).** The lower part of the table displays the correlation coefficients among the overall and subscale scores of the S-MARS-HA.

| *Descriptive statistics* | | | | | |
|---|---|---|---|---|---|
| | Overall score | Basic handling | Advanced handling | Adjustment | Aided listening |
| Median (p50) | 72.1 | 92.9 | 56.0 | 100.0 | 76.1 |
| Mean | 69.0 | 81.7 | 52.6 | 84.9 | 72.3 |
| Standard deviation | 15.5 | 25.0 | 26.7 | 22.8 | 20.1 |
| Minimum-maximum | 10.4–95.0 | 0–100 | 0–100 | 0–100 | 11.1–100 |
| Reliability (Cronbach's Alpha) | 0.88 | 0.90 | 0.72 | 0.84 | 0.89 |
| *Correlation coefficients among the overall and subscale scores of the S-MARS-HA* | | | | | |
| | Overall score | Basic handling | Advanced handling | Adjustment | Aided listening |
| Overall score | 1 | | | | |
| Basic handling | 0.74*** | 1 | | | |
| Advanced handling | 0.77*** | 0.66*** | 1 | | |
| Adjustment | 0.51*** | 0.35*** | 0.34*** | 1 | |
| Aided listening | 0.61*** | 0.13* | 0.15* | 0.22*** | 1 |

*p<0.05

***p<0.001

**Table 3. Descriptive statistics (median, 25th percentile and 75th percentile) and results of the confirmatory factor analysis (standardised factorial loads and determination coefficient ($R^2$)) of items from the Spanish version of the Measure of Audiologic Rehabilitation Self-Efficacy for Hearing Aids questionnaire (S-MARS-HA).**

| Items | Median (p25–p75) | Basic handling | Advanced handling | Adjustment | Aided listening | $R^2$ |
|---|---|---|---|---|---|---|
| 1. I can insert a battery into a hearing aid with ease (*Puedo colocar la pila en el audífono con facilidad*) | 100 (90–100) | 0.77*** | | | | 0.59*** |
| 2. I can remove a battery from a hearing aid with ease (*Puedo sacar la pila del audífono con facilidad*) | 100 (100–100) | 0.75*** | | | | 0.56*** |
| 3. I can tell a right hearing aid from a left hearing aid (*Puedo distinguir entre un audífono para el oído derecho y uno para el oído izquierdo*) | 100 (50–100) | 0.70*** | | | | 0.49*** |
| 4. I can insert hearing aids into my ears accurately (*Puedo colocar de manera correcta un audífono en mis oídos*) | 100 (100–100) | 0.78*** | | | | 0.61*** |
| 5. I can remove hearing aids from my ears with ease (*Puedo sacar un audífono de mis oídos con facilidad*) | 100 (100–100) | 0.76*** | | | | 0.58*** |
| 6. I can identify the different components of a particular hearing aid (i.e., microphone, battery door, vent, sound outlet, etc.).(*Puedo identificar las diferentes partes de un audífono (micrófono, portapilas, ventilación, parlante, etc.)*) | 60 (35–100) | 0.70*** | | | | 0.49*** |
| 7. I can operate all the controls on a particular hearing aid (knobs, switches, and/or remote control) appropriately (*Puedo manipular de manera correcta todos los controles de un audífono (botones, interruptores y/o control remoto)*) | 100 (50–100) | 0.71*** | | | | 0.50*** |
| 8. I can stop a hearing aid from squealing (*Puedo hacer que un audífono deje de chirriar*) | 60 (20–100) | | 0.68*** | | | 0.46*** |
| 9. I can troubleshoot a hearing aid when it stops working (Puedo resolver el problema de un audífono cuando deja de funcionar) | 80 (10–100) | | 0.62*** | | | 0.39*** |
| 10. I can clean and care for a hearing aid regularly (*Puedo limpiar y cuidar un audífono de manera frecuente*) | 100 (50–100) | | 0.55*** | | | 0.30*** |
| 11. I can name the make and model of a particular hearing aid (*Puedo nombrar la marca y el modelo de un determinado audífono*) | 10 (0–50) | | 0.49*** | | | 0.24*** |
| 12. I can name the battery size needed for a specific hearing aid (*Puedo nombrar el tamaño de una pila que usa un determinado un audífono*) | 50 (0–100) | | 0.55*** | | | 0.30*** |
| 13. I could get used to the sound quality of hearing aids (*Podría acostumbrarme a la calidad del sonido que entrega un audífono*) | 100 (70–100) | | | 0.82*** | | 0.67*** |
| 14. I could get used to how a hearing aid feels in my ear (*Podría acostumbrarme a la sensación de tener un audífono en mi oído*) | 100 (80–100) | | | 0.77*** | | 0.60*** |
| 15. I could get used to the sound of my own voice if I wore hearing aids (*Si usara audífonos, podría acostumbrarme al sonido de mi voz*) | 100 (80–100) | | | 0.83*** | | 0.69*** |
| 16. I could understand a one-on-one conversation in a quiet place if I wore hearing aids (*Si usara audífonos, podría entender una conversación con una persona en un lugar silencioso*) | 100 (90–100) | | | | 0.48*** | 0.23*** |
| 17. I could understand conversation in a small group in a quiet place if I wore hearing aids (*Si usara audífonos, podría entender una conversación con un grupo de persona en un lugar silencioso*) | 80 (60–100) | | | | 0.79*** | 0.62*** |
| 18. I could understand conversation on a regular telephone if I wore hearing aids (*Si usara audífonos podría entender una conversación por teléfono*) | 90 (50–100) | | | | 0.59*** | 0.35*** |
| 19. I could understand television if I wore hearing aids (*Si usara audífonos podría entender la televisión*) | 100 (80–100) | | | | 0.78*** | 0.61*** |
| 20. I could understand the speaker/lecturer at a meeting or presentation if I wore hearing aids (*Si usara audífonos podría entender a un expositor en una reunión o presentación*) | 85 (50–100) | | | | 0.78*** | 0.60*** |
| 21. I could understand a one-on-one conversation in a noisy place if I wore hearing aids (*Si usara audífonos, podría entender una conversación con una persona en un lugar ruidoso*) | 50 (40–80) | | | | 0.72*** | 0.52*** |
| 22. I could understand conversation in a small group while in a noisy place if I wore hearing aids (*Si usara audífonos, podría entender una conversación con un grupo de personas en un lugar ruidoso*) | 50 (20–70) | | | | 0.61*** | 0.38*** |

(*Continued*)

**Table 3.** (Continued)

| Items | Median (p25–p75) | Basic handling | Advanced handling | Adjustment | Aided listening | $R^2$ |
|---|---|---|---|---|---|---|
| 23. I could understand a public service announcement over the loudspeaker in a public building if I wore hearing aids (*Si usara audífono podría entender los avisos entregados a través de parlantes en lugares públicos*) | 70 (50–100) | | | | 0.67*** | 0.45*** |
| 24. I could understand conversation in a car if I wore hearing aids (*Si usara audífono, podría entender una conversación en un auto*) | 85 (50–100) | | | | 0.73*** | 0.53*** |

***$p<0.001$

## Association between educational level and hearing aid self-efficacy

**Overall score of the S-MARS-HA.** In the univariate model, years of formal education were positively and significantly (β = 0.57; 95% CI 0.15–0.99) associated with the overall S-MARS-HA scores (Table 4). This can be interpreted as the more years of formal education, the more significant increase in the S-MARS-HA overall score and thus on overall hearing aid self-efficacy. Other variables with a significant and positive association with the S-MARS-HA overall score in the univariate models included the participants' attitudes in using hearing aids (β = 1.65; 95% CI 0.73–2.48) and social support from friends or family members after getting the hearing aid (β = 2.85;

**Table 4.** Univariate and multivariate linear regression analyses for hearing aid self-efficacy (S-MARS-HA overall score).

| Independent variables | Univariate model (95%CI) | Multivariate model (95%CI)[a] |
|---|---|---|
| **Years of formal education** | 0.57 (0.15–0.99)** | 0.56 (0.02–1.10)* |
| **Pure-tone threshold average (PTA) in the fitted ear** | -0.25 (-0.45– -0.04)* | -0.21 (-0.43– -0.22)* |
| **Attitude to use the hearing aid** | 1.65 (0.73–2.48)*** | 1.17 (0.14–2.16)* |
| **Vision**[b] (very bad vision as a reference) | -1.65(-3.95–0.64) | – |
| **Joint problems** | -4.28(-7.98– -0.313)* | -3.09 (-7.33–0.99) |
| **Social support**[b] | | |
| Since getting your hearing aid, do you feel that you have had more or less support from your friends? (Much less support as a reference) | 2.85 (0.05–5.64)* | 1.33 (-2.78–5.26) |
| Since getting your hearing aid, are there more or fewer people who really care about you? (Many less people as a reference) | 2.64 (-0.81–5.97) | – |
| Since getting your hearing aid, do you feel that you have had more or less support from your family? (Much less support as a reference) | 4.42 (1.32–7.51)** | 2.56 (-2.02–7.18) |
| **Participation in social events since getting the hearing aid**[b] | | |
| Since getting your hearing aid, have you been able to participate in more or fewer social activities? (Participation in many more activities as a reference) | -3.48 (-6.05–0.80) | – |
| **Age (years)** | -0.23 (-0.55–0.16) | – |

[a] In the multivariate model, only variables significantly associated with the S-MARS-HA overall scores in the univariate models were included.

[b] Ordinal variables treated as continuous, thus the coefficient represents a change in hearing aid self-efficacy (i.e. S-MARS-HA score), passing from one category to another one.

* $p<0.05$

** $p<0.01$

*** $p<0.001$

95% CI 0.05–5.64 and β = 4.42; 95% CI 1.32–7.51). The presence of joint problems (β = -4.28; 95% CI -7.98– -0.313) and the PTA in the aided ear (β = -0.25; 95% CI -0.45– -0.04) were negative and significantly associated with the overall S-MARS-HA scores (Table 4).

In the multivariate model for the overall S-MARS-HA score, a significant association with the number of years of formal education was observed (β = 0.61; 95% CI 0.11–1.10). Attitudes in using hearing aids and the PTA in the aided ear were also significantly associated with the overall score of the S-MARS-HA.

**S-MARS-HA subscales.** The univariate models showed that both the *basic handling* and the *advanced handling* scores were significantly associated with years of formal education (β = 0.95; 95% CI 0.28–1.67 and β = 1.01; 95% CI 0.27–1.76 for each factor, respectively). In the multivariate model, the number of years of formal education was significantly associated with both *basic handling* and *advanced handling subscale* scores (β = 1.01; 95% CI 0.34–1.74 and β = 1.12; 95% CI 0.37–1.88 for each factor, respectively). This association was positive (i.e., the more years of formal education and the better the attitude in using hearing aids, the better the score on these two subscales) (Table 5). Attitudes in using the hearing aids were also significantly associated with both the *basic handling* and the *advanced handling* scores (β = 2.92; 95% CI 1.42–4.60 and β = 2.02; 95% CI 0.54–3.30 for each factor, respectively).

At both univariate and multivariate levels, the number of years of formal education was not significantly associated with the scores for the *adjustment* subscale of the S-MARS-HA. The

**Table 5. Univariate and multivariate linear regression analyses for the *basic handling* and *advanced handling* factors of the S-*MARS-HA* questionnaire.**

| Variables | Basic handling univariate model (95%CI) | Basic handling multivariate model (95%CI)[a] | Advanced handling Univariate model (95% CI) | Advanced handling Multivariate model (95% CI)[a] |
|---|---|---|---|---|
| Years of formal education | 0.95 (0.28–1.67)** | 1.01 (0.34–1.74)** | 1.01 (0.27–1.76)** | 1.12 (0.37–1.88)** |
| Pure-tone threshold average (PTA) in the fitted ear | -0.07 (-0.39–0.19) | – | -0.18 (-0.49–0.16) | – |
| Attitude to use the hearing aid | 3.06 (1.50–4.58)*** | 2.92 (1.42–4.60)*** | 2.18 (0.76–3.41)** | 2.02 (0.54–3.30)** |
| Vision[b] (Very bad vision as a reference) | -1.89 (-5.61–1.98) | – | 1.80 (-2.05–5.77) | – |
| Joint problems | -6.30 (-12.37– -0.13)* | -4.19 (-9.82–1.65) | -6.82 (-13.32– -0.16)* | -5.04 (-11.56–1.43) |
| Social support[b] | | | | |
| Since getting your hearing aid, do you feel that you have had more or less support from your friends? (Much less support as a reference) | 2.14 (-2.58–6.54) | – | 4.40 (-1.27–9.85) | – |
| Since getting your hearing aid, are there more or fewer people who really care about you? (Many less people as a reference) | 3.22 (-1.93–8.24) | – | 4.16 (-2.55–10.68) | – |
| Since getting your hearing aid, do you feel that you have had more or less support from your family? (Much less support as a reference) | 4.77 (-0.08–9.62) | – | 6.45 (0.49–12.17)* | 2.02 (-0.54–3.30) |
| Participation in social events since getting the hearing aid[b] | | | | |
| Since getting your hearing aid have you been able to participate in more or fewer social activities? (Participation in many more activities as a reference) | -2.08 (-6.25–5.91) | – | -2.41 (-7.51–3.65) | – |
| Age (years) | -0.14 (-0.64–0.39) | – | -0.50 (-1.02–0.05) | – |

[a] In the multivariate model, only variables significantly associated with the S-MARS-HA overall scores in the univariate models were included.

[b] Ordinal variables treated as continuous, thus the coefficient represents a change in hearing aid self-efficacy (i.e. S-MARS-HA score), passing from one category to another one.

* p<0.05

** p<0.01

*** p<0.001

**Table 6. Univariate and multivariate linear regression for the *adjustment* and *aided listening* factors of the *S-MARS-HA* questionnaire.**

| Variables | Adjustment univariate model (95%CI) | Adjustment multivariate model (95%CI)[a] | Aided Listening univariate model (95% CI) | Aided listening multivariate model (95% CI)[a] |
|---|---|---|---|---|
| **Years of formal education** | 0.38 (-0.99–0.24) | – | 0.31 (-0.25–0.85) | – |
| **Pure-tone threshold average (PTA) in the fitted ear** | -0.06 (-0.21–0.31) | – | -0.58 (-0.82– -0.34)*** | -0.55 (-0.79– -0.32)*** |
| **Attitude to use the hearing aid** | 3.63 (2.23–4.94)*** | 3.49 (2.11–4.88)*** | -0.55 (-1.28–0.29) | – |
| **Vision[b] (Very bad vision as a reference)** | -3.67 (-7.28– -0.37)* | -2.06 (-5.43–1.21) | -2.16 (-5.38–1.03) | – |
| **Joint problems** | -3.55 (-9.17–2.07) | – | -2.21 (-7.13–2.73) | – |
| **Social support[b]** | | | | |
| Since getting your hearing aid, do you feel that you have had more or less support from your friends? (Much more support as a reference) | 4.21 (0.50–8.12)* | 2.66 (-0.75–6.57) | 2.25 (-2.01–6.65) | – |
| Since getting your hearing aid, are there more or fewer people who really care about you? (Many more people as a reference) | 2.55 (-2.56–7.14) | – | 2.74 (-1.83–6.95) | – |
| Since getting your hearing aid, do you feel that you have had more or less support from your family? (Much more support as a reference) | 2.70 (-1.98–7.36) | – | 4.11 (-0.19–8.32) | – |
| **Participation in social events since getting the hearing aid[b]** | | | | |
| Since getting your hearing aid, have you been able to participate in more or fewer social activities? (Participation in many more activities as a reference) | -2.31 (-7.05–2.75) | – | -6.42 (-11.23– -2.70)** | -6.90 (-12.73– -2.13)* |
| **Age (years)** | -0.04 (-0.54–0.44) | – | -0.20 (-1.02–0.05) | – |

[a] In the multivariate model, only variables significantly associated with the S-MARS-HA overall scores in the univariate models were included.

[b] Ordinal variables treated as continuous, thus the coefficient represents a change in hearing aid self-efficacy (i.e. S-MARS-HA score), passing from one category to another one.

* p<0.05

** p<0.01

*** p<0.001

same is true for the possible association between the number of years of formal education and the *aided listening* subscale scores of the S-MARS-HA. However, in the multivariate model, the attitude in using hearing aids was the only covariate that was significantly associated with the *adjustment* subscale of the S-MARS-HA (β = 3.49; 95% CI 2.11–4.88) (Table 6). Both at univariate and multivariate levels, the *aided listening* subscale scores were negatively and significantly associated with changes in social participation after getting the hearing aid. The scores in this subscale decreased as older adults participated in fewer social activities (β = -6.90; 95% CI -12.73– -2.13). The PTA in the aided ear (β = -0.55; 95% CI -0.79– -0.32) was also significantly associated with the scores for this subscale in both univariate and multivariate models.

## Association between hearing aid self-efficacy and hearing aid adherence

A significant association between hearing aid self-efficacy and adherence to hearing aid use was found. Such an association was found between the IOI-HA question and both S-MARS-HA overall score (OR = 1.04; 95% CI 1.03–1.06; *p* < 0.001) and *basic handling* (OR = 1.03; 95% CI 1.02–1.04; *p* < 0.001), *advanced handling* (OR = 1.03; 95% CI 1.02–1.04; *p* < 0.001), and *adjustment* (OR = 1.04; 95% CI 1.02–1.05; *p* < 0.001) subscales of the S-MARS-HA. There was no significant association between the *aided listening* S-MARS-HA subscale and *adherence to hearing aid use* (OR = 0.99; 95% CI 0.98–1.00; *p* = 0.10). Fig 1 shows that as the overall score for the S-MARS-HA increases, the probability of responding that they "never" use hearing aids

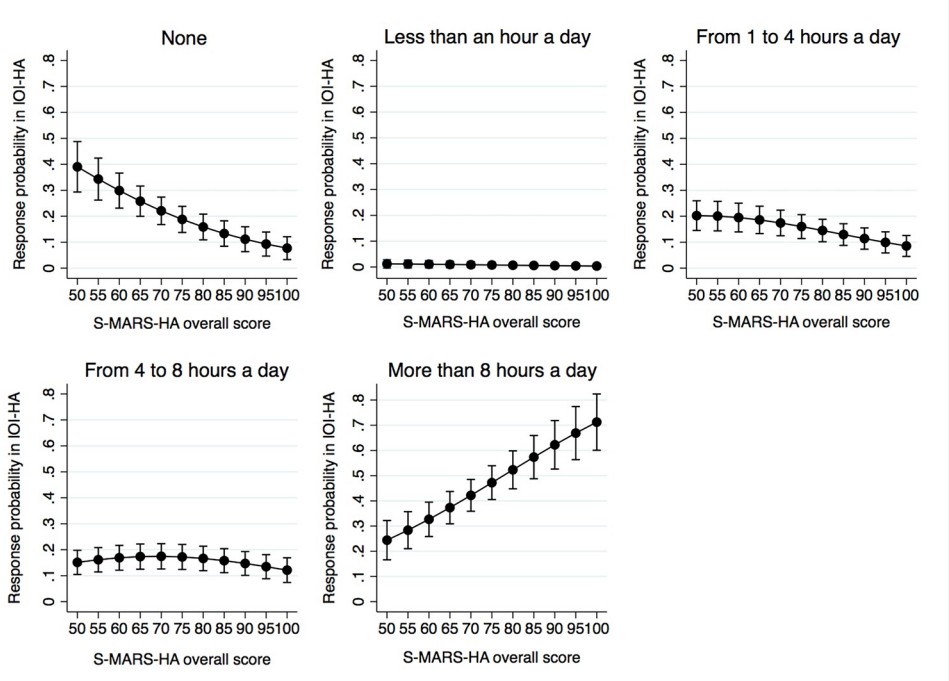

**Fig 1. Association between the scores for the IOI-HA (i.e. hearing aid adherence) and the overall scores for the S-MARS-HA (hearing aid self-efficacy).** IOI-HA question 1: Think about how much you used your present hearing aid(s) over the past two weeks. On an average day, how many hours did you use the hearing aid(s)?

decreases. The probability of using a hearing aid for more than eight hours increased as the S-MARS-HA overall score did (Fig 1). This pattern was observed in all the factors associated with adherence to hearing aid use.

## Discussion

### Validity and reliability of the S-MARS-HA

The confirmatory factor model in which the items were grouped into four factors like in the original version of the MARS-HA [10] had good goodness-of-fit indicators. However, the error variances were freely estimated for the relationship between items 1 and 2, 4 and 5, and 21 and 22. The fact that these pairs of items were related between them is likely explained by an association with a variable other than the factor with which they loaded. According to Byrne such a covariance between items can result from overlapping item content, and this is to be expected as both items involve similar motor skills [39]. Being able to put in and take out the hearing aid battery—skills evaluated in items 1 and 2—involves similar motor skills. The former is valid for items 4 and 5 (being able to put the hearing aid in and take it out). Items 21 and 22 involve complex acoustic situations, which are associated with the technological features of hearing aids. Participants were fitted with hearing aids that have neither directional microphones nor algorithms for noise reduction.

The reliability of the S-MARS-HA is high and is similar to that obtained with the original version. It is also similar to the French version in Quebec [26], which included older adult hearing aid users (from 66 to 88 years old) and for which a similar cross-cultural adaptation process was used. In the adaptation of the MARS-HA into Canadian French, two initial translations into French were obtained independently, and then a committee of experts reviewed

both translations to generate a single instrument in French. The latter was then back translated into English and compared to the original English version of the instrument.

The correlation among the subscales and the overall score is strong, although correlations between certain subscales were weak. This was the case for the correlation between the *aided listening* and the *basic handling* subscales along with the correlation between the *aided listening* and the *advanced handling* subscales. These weak correlations may be explained by the fact that the *aided listening subscale* had relatively lower scores than the other subscales and that very few older adults increased their participation in social activities after being fitted with the hearing aid (only 2.4% of the sample). Another explanation could be that the MARS-HA questionnaire includes four different constructs in which different skills are involved. For example, motor skills are associated with the *basic* and *advanced handling subscales*, and communicative skills are associated with the aided listening and Adjustment subscales.

## Association between educational level and hearing aid self-efficacy

The primary aim of this study was to determine a possible association between educational level and hearing aid self-efficacy in a sample of Chilean older adults. This was because of the fact that, as opposed to developed countries, in Latin American countries, educational level is not uniformly distributed among older adults [13]. In this study, we found that years of formal education significantly predicted hearing aid self-efficacy. These results indicate that older adults with higher educational levels exhibited higher hearing aid self-efficacy.

This is the first study investigating the effect of educational level (i.e., years of formal education) on hearing aid self-efficacy in older adults. We hypothesize that such an association may be explained by the readability of the written materials (i.e., hearing aid user's guide) provided to the patients and their capacity to understand such materials based on their educational level. Caposecco et al., [14] investigated the content, design, and readability of hearing aid user manuals and found that they were inadequate (i.e., a readability higher than that of ninth-grade level). In the case of the present sample of participants, the average number of years spent in education was low (8.9 years, equivalent to full primary education). This is much lower than in developed countries such as the USA, where the population aged 65 and over who completed high school is 84.4% [40]. In this study, we did not systematically investigate the readability of the hearing aid user manuals, but we strongly believe they were not designed to be clearly understood by the population of older adults attending public hospitals in Chile. Supporting evidence for this hypothesis is the fact that educational level was significantly associated only with *basic* and *advanced handling* subscales of the S-MARS-HA. Both aspects of hearing aid self-efficacy strongly depend on the user's understanding on how to operate the device. Such an understanding is likely obtained from hearing aid user manuals. Additionally, handling the hearing aid can be learned from follow-up sessions after the hearing aid is fitted. In the case of the public system in Chile, where this sample of older adults came from, older adults are provided with three sessions after the hearing aid is fitted. However, they mainly cover hearing aid fitting aspects, and the clinical guidelines issued by the Chilean Ministry of Health do not specifically suggest the inclusion of educational aspects about how to handle the hearing aid. Thus, further research should investigate the readability of the hearing aid user manuals provided to older adults in Latin American countries. In addition, strategies to improve older adults' understanding about how to operate their hearing aids should be developed for this patient population. Such strategies should be incorporated in follow-up sessions and should be tailor-made according to patients' educational level.

The attitude in using a hearing aid was a covariate significantly associated with the S-MARS-HA overall score. This means that older adults who presented with a more positive attitude

to a hearing aid wore it for more hours during the day. We hypothesize that a positive attitude to hearing aids was a key variable leading to older adults developing hearing aid self-efficacy throughout time. We speculate that older adults with a positive attitude to hearing aids kept trying to wear the device for more periods of time, even if they had difficulties that may affect handling the device such as visual and joint problems. Wearing the device for longer periods of time ultimately allowed older adults to develop hearing aid self-efficacy owing to their experience with the device. This is in agreement with Meyer et al. [9] who found that the length of hearing aid use was a significant predictor of the advanced handling subscale of the MAR-S-HA. Similarly, Hickson et al. [8] showed that people with positive attitudes towards hearing aids and greater confidence to manage advanced features of a hearing aid were more likely to report a successful outcome with hearing aids. In this study, when the multivariate regression model was adjusted for the attitude in using a hearing aid, both vision and joint problems were no longer significantly associated with hearing aid self-efficacy, even if they both were significantly associated with this outcome measure in the univariate regression models. In the present study on hearing aid self-efficacy, attitudes in using hearing aids had an independent effect from the participants' educational level. This is because attitudes to hearing aids and educational levels were not significantly associated in the regression models.

The PTA in the fitted ear was another covariate significantly associated with hearing aid self-efficacy. This covariate significantly predicted the S-MARS-HA overall score and the *aided listening* subscale score of the S-MARS-HA. Therefore, we hypothesize that the effect of PTA on hearing aid self-efficacy is basically driven by the effect of audibility on the person's capacity to understand other people talking while wearing a hearing aid. This is in agreement with previous studies [9, 11] where aided listening as a measure for hearing aid self-efficacy was negative and significantly associated with the degree of hearing loss in participants' worse ear.

Concerning specific aspects of hearing aid self-efficacy (i.e., S-MARS-HA subscales), *basic handling* and *advanced handling* were both significantly predicted by the number of years of formal education. As discussed above, we strongly believe that such an association owes to the readability of the hearing aid user manuals which depends on people's educational level. The attitude in using a hearing aid was a covariate significantly associated with *basic handling*, *advanced handling*, and (the only covariate associated with it) *adjustment* subscales of the S-MARS-HA. The discussion provided above about the effect of the participants' attitudes to hearing aids on the overall score of the S-MARS-HA may well explain the associations between the former variable and the S-MARS-HA subscale scores. The PTA and participation in social activities after getting the hearing aid were both negative and significantly associated with the *aided listening* subscale score of the S-MARS-HA. We believe that participating in more social activities allows people to experience a richer variety of listening environments [41], thus providing a sort of listening training with the hearing aid. So the more social activities the older adult participates in while wearing the hearing aid, the better their performance for listening activities in daily life, which should be reflected by better scores for the *aided listening* subscale of the S-MARS-HA. The PTA in the fitted ear directly relates with access to speech sounds. Participants with poorer hearing thresholds may still present with more difficulties accessing speech sounds than participants with better hearing thresholds [42], especially in the high-frequency range, while wearing a hearing aid. This may well explain the association between the unaided PTA and the *aided listening* subscale of the S-MARS-HA.

Years of formal education, attitudes in using hearing aids, and changes in social participation after getting a hearing aid were not investigated in the previous study by Meyer et al. [9] about the barriers and facilitators for hearing aid self-efficacy. In that study, the scores for the *basic handling* subscale of the MARS-HA were significantly associated with social support. Such an association was not observed in the present study. The differences between our results

and Meyer et al. [9] may be explained by the way the variable hearing aid self-efficacy was treated. Meyer et al. [9] dichotomized the variable with a cutoff point of 80% to determine that hearing aid self-efficacy was adequate. In our study, hearing aid self-efficacy was always treated as a continuous variable. Further studies are needed to better understand the effect of social support on hearing aid self-efficacy in older adults.

In summary, this study shows that educational level is significantly associated with hearing aid self-efficacy in Chilean older adults. Such an association has not been previously reported based on samples of older adults from developed countries (e.g., Meyer et al., [9]) where older adults' educational level tend to be higher and equally distributed than developing countries. Thus, we strongly suggest that this demographic variable should be considered when designing and delivering aural rehabilitation programs to older adults, especially from developing economies. In addition, we strongly believe that clinicians working in Latin America and other world regions where the educational level of older adults is low should develop easy-to-read instructions on how to operate hearing aids. Covariates significantly associated with hearing aid self-efficacy included attitudes in using hearing aids and the PTA in the fitted ear. The PTA has also been found to be associated with hearing aid self-efficacy in older adults in previous studies conducted in developed countries [10, 15].

## Association between hearing aid self-efficacy and hearing aid adherence

There was a significant association between hearing aid self-efficacy and adherence to hearing aid use (IOI-HA question). This was observed for the S-MARS-HA overall score and for the *basic handling*, *advanced handling*, and *adjustment* subscales. Greater hearing aid self-efficacy was significantly associated with a higher score in the IOI-HA question (i.e., greater adherence). This is similar to the findings reported by Hickson et al. [8]. The authors found significant differences for all subscale scores of the MARS-HA between successful and not successful older adult hearing aid users. The latter was determined based on participants' responses for questions 1 and 2 of the IOI-HA. Similar results were also reported by Meyer et al. [9]. Therefore, based on this and previous studies, it may be concluded that hearing aid self-efficacy significantly predicts hearing aid adherence in older adults, independent on whether they are monaurally (i.e., as in this study) or binaurally fitted (e.g., Hickson et al., [8]; Meyer et al., [9]). It should be noted, however, that all studies (including the present one) have investigated hearing aid self-efficacy and adherence to the use of hearing aid in a cross-sectional manner at the same point in time. Therefore, it cannot be assumed that older adults who may present adequate hearing aid self-efficacy shortly after the hearing aid is fitted will utilize the hearing aid in the future. Further studies should investigate how hearing aid self-efficacy at an initial stage of the aural rehabilitation can predict future use of hearing aids.

## Limitations and recommendations for further studies

One of the main limitations of this study is that the levels of hearing aid self-efficacy can change over time, and that was not investigated in the present study because of the cross-sectional design of the research. Therefore, a reverse causation bias may partially explain the association between hearing aid self-efficacy and hearing aid adherence. Future studies should longitudinally investigate changes (e.g., improvements) in hearing aid self-efficacy and its association with hearing aid adherence. Another limitation relates to the construct validity of the S-MARS-HA. Although most of the questions of the S-MARS-HA showed a high $R^2$, questions 11 and 16 showed a rather low $R^2$. Future studies should evaluate the relevance of these items. In addition, it cannot be assumed that all of the MARS-HA questions had the same capacity to discriminate between good and poor self-efficacy. The discrimination capacity of each item

can be formally evaluated using the item response theory, a psychometric model that has recently been incorporated into health research [43]. Future studies should evaluate hearing aid self-efficacy by selecting the items from the MARS-HA that have the greatest capacity to differentiate between older adults with good and poor hearing aid self-efficacy. By doing this, a shorter version of the MARS-HA can be developed, making it easier to use in clinical contexts. Another possible limitation relates to the use of a self-report measure of adherence rather than hearing aid data-logging. Data logging could not be accessed in this study because data collection was carried out at participants' homes. However, Laplante-Lévesque et al. [44] reported minor differences for adherence to hearing aids between the use hearing aid data-logging and self-report questionnaires, with an over report of daily hearing aid use of 1.2 hours. Therefore, it is very likely that the use of a self-report questionnaire did not affect the results of this study.

## Conclusions

Educational level was significantly associated with hearing aid self-efficacy in a sample of experienced older adult hearing aid users from Chile. Specifically, the number of years of formal education significantly predicted the overall score of the S-MARS-HA and the *basic handling* and *advanced handling* subscale scores of the S-MARS-HA. We hypothesize that educational level has a positive effect on the person's capacity to understand the hearing aid user manuals and ultimately on their capacity to handle the hearing aid. To the best of our knowledge, this is the first study reporting an association between educational level and hearing aid self-efficacy in older adults. This may be explained by the fact that in developing countries such as Chile, older adults' educational levels are lower and unequally distributed compared to developed countries. Therefore, we strongly suggest that educational level should be taken into consideration when designing and delivering aural rehabilitation to older adults, especially in developing nations. Similar to other studies, attitudes in using hearing aids and audibility were significantly associated with hearing aid self-efficacy in older adults. Finally, we observed a significant association between hearing aid self-efficacy and hearing aid adherence. This finding has been previously reported with samples of older adults from developed countries. Thus, it can be concluded that hearing aid self-efficacy is a key element in the process of becoming a successful hearing aid user. Further studies should investigate how hearing aid self-efficacy changes over time and its impact on hearing aid adherence in older adults.

## Supporting information

**S1 Annex. Spanish version of the Measure of Audiologic Rehabilitation Self-Efficacy for Hearing Aids questionnaire (S-MARS-HA).**
(DOCX)

**S1 Datasetdta. Data used in all analyses.**
(ZIP)

## Acknowledgments

The authors would like to thank all research participants for their time and interest in this study.

## Author Contributions

**Conceptualization:** Eduardo Fuentes-López, Adrian Fuente, Gonzalo Valdivia.

**Data curation:** Eduardo Fuentes-López.

**Formal analysis:** Eduardo Fuentes-López.

**Funding acquisition:** Eduardo Fuentes-López, Adrian Fuente, Gonzalo Valdivia.

**Investigation:** Eduardo Fuentes-López, Adrian Fuente, Gonzalo Valdivia.

**Methodology:** Eduardo Fuentes-López, Adrian Fuente, Gonzalo Valdivia.

**Project administration:** Eduardo Fuentes-López, Gonzalo Valdivia, Manuel Luna-Monsalve.

**Resources:** Manuel Luna-Monsalve.

**Software:** Eduardo Fuentes-López.

**Supervision:** Manuel Luna-Monsalve.

**Validation:** Eduardo Fuentes-López, Adrian Fuente, Gonzalo Valdivia, Manuel Luna-Monsalve.

**Visualization:** Eduardo Fuentes-López, Adrian Fuente.

**Writing – original draft:** Eduardo Fuentes-López, Adrian Fuente.

**Writing – review & editing:** Eduardo Fuentes-López, Adrian Fuente.

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
