## [Decision Letter · Decision Letter 0]

25 Sep 2019

PONE-D-19-19089

Does Educational Level Predict Hearing Aid Self-Efficacy in Experienced Older Adult Hearing Aid Users from Latin America?

PLOS ONE

Dear Dr. Fuentes-López,

Thank you for submitting your manuscript to PLOS ONE. After careful consideration, we feel that it has merit but does not fully meet PLOS ONE’s publication criteria as it currently stands. Therefore, we invite you to submit a revised version of the manuscript that addresses the points raised during the review process.

Reviewers expressed a positive opinion on the manuscript, suggesting only minor changes. I therefore recommend the Authors to address all the Reviewers’ requests to make the manuscript suitable for publication.

We would appreciate receiving your revised manuscript by Nov 09 2019 11:59PM. To enhance the reproducibility of your results, we recommend that if applicable you deposit your laboratory protocols in protocols.io, where a protocol can be assigned its own identifier (DOI) such that it can be cited independently in the future. For instructions see: http://journals.plos.org/plosone/s/submission-guidelines#loc-laboratory-protocols

We look forward to receiving your revised manuscript.

Kind regards,

Stefano Federici, Ph.D.

Academic Editor

PLOS ONE

Journal Requirements:

Additional Editor Comments (if provided):

Reviewers expressed a positive opinion on the manuscript, suggesting only minor changes. I therefore recommend the Authors to address all the Reviewers’ requests to make the manuscript suitable for publication.

Reviewers' comments:

Reviewer's Responses to Questions

**Comments to the Author**

1. Is the manuscript technically sound, and do the data support the conclusions?

Reviewer #1: Yes

Reviewer #2: Yes

2. Has the statistical analysis been performed appropriately and rigorously? 

Reviewer #1: Yes

Reviewer #2: Yes

3. Have the authors made all data underlying the findings in their manuscript fully available?

Reviewer #1: Yes

Reviewer #2: No

4. Is the manuscript presented in an intelligible fashion and written in standard English?

Reviewer #1: Yes

Reviewer #2: No

5. Review Comments to the Author

Reviewer #1: Tittle:

Does Educational Level Predict Hearing Aid Self-Efficacy in Experienced Older Adult Hearing Aid Users from Latin America?

Version: August 30th, 2019.

PLOS ONE

Reviewer: Fernando Gomez

This is a cross sectional observational study in a convenience sample from an urban hearing impairment outpatient program of Governmental hospital in Santiago (Chile). This paper has three components, one is the validity of a questionnaire used to measure hearing aid self-efficacy: the Spanish version of Measure of Audiologic Rehabilitation Self-Efficacy for Hearing Aids (S- MARS-HA); another second component about the role of educational level and other factors as attitudes in using hearing aids and the pure-tone average, (PTA) in the fitted ear as important factor of hearing aid self efficacy and, finally the third component of this manuscript explores the association between hearing aid self-efficacy with adherence to hearing aid use. The topic of the manuscript is appropriate for the Journal. It could be of interest to investigators and clinicians. Minor essential revisions are necessary.

Tittle:

The title only described one component of the paper, educational level and hearing self-efficacy; any alternative to wide all topics included in the manuscript? However, the title is consistent with the presented problem and reflects the main message of the study.

Abstract:

Abstract: Concise and specific.

Main objective of the study is presented.

The statistical methods used to prove the hypothesis is mentioned.

Conclusion highlights the contribution of this work.

Introduction:

What is the rationale of relationship between educational level and hearing aid self-efficacy? Please provide in introduction section short information about it. Only comments about suitability of hearing aid user guides for older people is founded. If not is possible at least the relationship between education and wear hearing aids.

About educational level and hearing aid adoption, please check:

Popelka M.M., Cruickshanks K.J., Wiley T.L., Tweed T.S., Klein B.E. et al. 1998. Low prevalence of hearing aid use among older adults with hearing loss: The Epidemiology of Hearing Loss Study. J Am Geriatr Soc , 46, 1075 – 1078

Tomita M., Mann W.C. & Welch T.R. 2001. Use of assistive devices to address hearing impairment by older persons with disabilities. Int J Rehabil Res , 24, 279 – 289.

Kochkin S. 2009. MarkeTrak VIII: 25-year trends in the hearing health market. The Hearing Review , 16, 12 – 31.

Material and methods:

Sufficient details about the process are provided.

Statistical analyses used are appropriate.

The methods are appropriate and well described.

Results:

Information is clearly provided.

One figure and six tables are clear and well designed.

Supporting information in annex including Spanish version of tool is provided.

Discussion:

Discussion should no include results, for example about validity and reliability of the S-MARS-HA. Please revise first paragraph of discussion section.

Please check second paragraph about comparison of reliability of S-MARS-HA with literature, in this case with Canadian-French version of Stigma Consciousness Questionnaire (SCQ), it is necessary provide more information, rather than similar age in both studies.

What is the reference supporting the explanation of weak correlations in several subscales of the tool?

To support their findings authors should insist in relationship with self-efficacy and education in previous original paper Bandura`s self-efficacy thesis, see references of Meyer C, Hickson L, Fletcher A. Identifying the barriers and facilitators to optimal hearing aid self-efficacy. Int J Audiol. 2014; 53(suppl 1): S28-S37, and internal locus of control and educational level.

Why authors shown results in discussion section?, see “wear of hearing aid”, page 29-30. Time of wearing device is an excellent hypothesis for positive attitude, please reinforce it with appropriate reference. Please check in this reference provided by authors: Ng JH, Loke AY. Determinants of hearing-aid adoption and use among the elderly: a systematic review. Int J Audiol. 2015; 54(5): 291-300.

In the same way, paragraph about PTA and participation in social activities as a factor for better adherence to hearing aid deserve other reference.

References than support several comments are necessary, for example page 29 paragraph about “ handling hearing aid and educational level or page 29 and 30 about attitudes to hearing aids, please see excellent references provided about barriers to help-seeking for hearing aid adoption.

References:

There were 32 and all are appropriate.

Thanks for letting me review this manuscript.

This could be a nice paper.

Level of interest: An article whose findings are important to those with closely related research interests.

Quality of written English: Well.

Statistical review: No.

Declaration of competing interests:

I declare that I have no competing interest.

Reviewer #2: The answer of "no" to Question 4 is only referring to a few minor grammatical issues flagged in the attachment. This is a well-written paper overall, but according to the review instructions I would choose "yes" only if I had no revisions at all.

The answer of "no" to Question 3: The instructions state that the data should be included as a supplemental file or publicly available. I was not able to find this in the supplemental file that was included with this submission, nor could I find a website that makes the data available - my apologies if I missed it. I do see your statement that the data will be made available - I will leave it to the editor if this can simply be added as a revision to this paper.

6. PLOS authors have the option to publish the peer review history of their article (what does this mean?). If published, this will include your full peer review and any attached files.

Reviewer #1: No

Reviewer #2: No

---

## [Author Response · Author response to Decision Letter 0]

3 Nov 2019

The following changes have been made according to the specific comments previously received:

Reviewer #1: 

This is a cross sectional observational study in a convenience sample from an urban hearing impairment outpatient program of Governmental hospital in Santiago (Chile). This paper has three components, one is the validity of a questionnaire used to measure hearing aid self-efficacy: the Spanish version of Measure of Audiologic Rehabilitation Self-Efficacy for Hearing Aids (S- MARS-HA); another second component about the role of educational level and other factors as attitudes in using hearing aids and the pure-tone average, (PTA) in the fitted ear as important factor of hearing aid self efficacy and, finally the third component of this manuscript explores the association between hearing aid self-efficacy with adherence to hearing aid use. The topic of the manuscript is appropriate for the Journal. It could be of interest to investigators and clinicians. Minor essential revisions are necessary.

Tittle:

The title only described one component of the paper, educational level and hearing self-efficacy; any alternative to wide all topics included in the manuscript? However, the title is consistent with the presented problem and reflects the main message of the study.

R. According a suggestion from the second reviewer (see below) the title was modified with the aim of including the validation process of the questionnaire: Does Educational Level Predict Hearing Aid Self-Efficacy in Experienced Older Adult Hearing Aid Users from Latin America? Validation process of the Spanish version of the MARS-HA questionnaire.

Abstract:

Abstract: Concise and specific.

Main objective of the study is presented.

The statistical methods used to prove the hypothesis is mentioned.

Conclusion highlights the contribution of this work.

Introduction:

What is the rationale of relationship between educational level and hearing aid self-efficacy? Please provide in introduction section short information about it. Only comments about suitability of hearing aid user guides for older people is founded. If not is possible at least the relationship between education and wear hearing aids.

R. We strongly believe that the effect of educational level on the hearing aid self-efficacy is mediated mainly by suitability of hearing aid user guides for older people. As far as we know, this could be the only variable taken into account by other previous studies. However, as suggested by the reviewer, we have further elaborated this hypothesis. Please see page 4, lines 23-24; page 5, lines 1-14, and lines 18-22 of the revised version of the manuscript. 

Material and methods:

Sufficient details about the process are provided.

Statistical analyses used are appropriate.

The methods are appropriate and well described.

Results:

Information is clearly provided.

One figure and six tables are clear and well designed.

Supporting information in annex including Spanish version of tool is provided.

Discussion:

Discussion should no include results, for example about validity and reliability of the S-MARS-HA. Please revise first paragraph of discussion section.

R. As suggested by the reviewer, we excluded from this paragraph the information about actual results. However, some information about the construct validity was not removed in order to support the explanation about the error variances estimation. Error variances estimation refers to the relationship between items that are likely explained by their association with a variable other than the factor they loaded. According to Byrne (2013) such covariance between items can result from overlapping item content. Therefore, the information about the overlapping item content was kept (see page 28, lines 4-11).

References

- Byrne BM. Structural equation modeling with Mplus: Basic concepts, applications, and programming. Routledge; 2013.

Please check second paragraph about comparison of reliability of S-MARS-HA with literature, in this case with Canadian-French version of Stigma Consciousness Questionnaire (SCQ), it is necessary provide more information, rather than similar age in both studies.

R. As suggested by the reviewer, we have now extended this discussion. See page 28, lines 20-23 and page 29, lines 1-2).

What is the reference supporting the explanation of weak correlations in several subscales of the tool?

R. In the current study, the aided listening sub-scale concentrates the weak correlations. This is similar to the previous validation of MARS-HA questionnaire by West & Smith (2007) where the correlation between Aided listening and Advanced handling sub-scales was weak (r=0.29). In addition, the correlation between Aided listening and basic handling sub-scales was moderate (r=0.46). Although West & Smith did not provide an explanation about these findings, we believe that these correlations may be explained by the fact that the aided listening subscale had relatively lower scores than the other subscales. This could be related to the fact that very few older adults increased their participation in social activities after being fitted with the hearing aid (only 2.4% of the sample), exposing themselves poorly to communicate with others (skills included into Aided listening sub-scale). Another explanation could be that MARS-HA questionnaire includes four different constructs in which different skills are involved. In the case of the Basic and Advanced handling motor skills are involved. On the contrary, for the Aided listening and Adjustment subscales underlie communicative skills. We added more information for explaining the weak correlations in the discussion section (see page 29, lines 11-14).

Reference

- West RL, Smith SL. Development of a hearing aid self-efficacy questionnaire. Int J Audiol. 2007; 46(12): 759-71.

To support their findings authors should insist in relationship with self-efficacy and education in previous original paper Bandura`s self-efficacy thesis, see references of Meyer C, Hickson L, Fletcher A. Identifying the barriers and facilitators to optimal hearing aid self-efficacy. Int J Audiol. 2014; 53(suppl 1): S28-S37, and internal locus of control and educational level.

R. Meyer et al. (2014) cited Smith & West (2006), an article where they mentioned the sources of self-efficacy beliefs proposed by Bandura (1977, 1986, 1997). According to Bandura, individuals formulate self-efficacy beliefs primarily from four sources of information: (a) enactive mastery experiences, (b) vicarious experience, (c) verbal persuasion, and (d) physiological and affective states. Smith & West (2006) provided examples related to audiological rehabilitation for adults with hearing loss in which it is possible to identify the effect of educational level on some of the sources of self-efficacy beliefs previously mentioned. It is possible that the effect of educational level on self-efficacy mainly occurs through “Mastery Experience” and “Verbal persuasion” sources of information. One strategy to increase the “Mastery Experience” is to divide new skills into smaller steps as it is incorporated in the hearing aid user guides for older people, whereas “Verbal persuasion” can be increased by giving appropriate feedback or providing pedagogic materials, such as are provided in clinical controls. In both cases, having a higher educational level would facilitate the aforementioned strategies to increase “Mastery Experience” and “Verbal persuasion” sources of information. We added more information at this respect in the introduction section (see page 4, lines 23-24 and page 5, lines 1-14).

References

− Bandura A. Self-efficacy: Toward a unifying theory of behavior change. Psychological Review. 1987; 84: 191–192.

− Bandura A. Social foundations of thought and action: A social cognitive theory. Englewood Cliffs, NJ: Prentice-Hall. 1986.

− Bandura A. Self efficacy: The exercise of control. New York: Freeman. 1997.

− Smith SL, West RL. The application of self-efficacy principles to audiologic rehabilitation: a tutorial. Am J Audiol. 2006; 15(1): 46-56. 

Why authors shown results in discussion section?

R. The odds ratio reported was eliminated from this current version of the manuscript.

See “wear of hearing aid”, page 29-30. Time of wearing device is an excellent hypothesis for positive attitude, please reinforce it with appropriate reference. Please check in this reference provided by authors: Ng JH, Loke AY. Determinants of hearing-aid adoption and use among the elderly: a systematic review. Int J Audiol. 2015; 54(5): 291-300.

R. References were added according to the reviewer’s comment. The studies carried out by Meyer et al. and Hickson et al. included information about the relationship between attitudes, hearing aids self-efficacy and adherence. Meyer et al. found that the length of hearing aid use was a significant predictor of advanced handling subscale. Similarly, Hickson et al. observed that older adults who possessed both a positive attitude towards hearing aids and greater confidence in their ability to manage the more advanced features of a hearing aid were more likely to report a successful outcome with hearing aids (see page 31, lines 15-20).

In the same way, paragraph about PTA and participation in social activities as a factor for better adherence to hearing aid deserve other reference.

R. Results from studies carried out by Heffernan et al. (2016), and Wu & Bentler (2012) were incorporated. In Heffernan et al. (2016), UK older adults with mild-moderate hearing loss reported activity limitations and participation restrictions (i.e. conversing with the others). Wu & Bentler (2012) recruited North American adults with bilateral hearing impairment aged 40–88 years. Older adults (> 65 years) had less active social lifestyles (smaller social networks) requiring fewer demands on hearing than younger adults (< 65 years), matched by hearing thresholds. These studies support our statement that participating in more social activities allows people to experience a richer variety of listening environments, thus providing a sort of listening training with the hearing aid (see page 33, lines 1-11).

References

- Heffernan E, Coulson NS, Henshaw H, Barry JG, Ferguson M. Understanding the psychosocial experiences of adults with mild-moderate hearing loss: An application of Leventhal's self-regulatory model. Int J Audiol. 2016; 55 Suppl 3: S3-S12. 

- Wu YH, Bentler RA. Do older adults have social lifestyles that place fewer demands on hearing? J Am Acad Audiol. 2012; 23(9): 697-711.

References than support several comments are necessary, for example page 29 paragraph about “handling hearing aid and educational level or page 29 and 30 about attitudes to hearing aids, please see excellent references provided about barriers to help-seeking for hearing aid adoption.

R. References were provided accordingly.

References:

There were 32 and all are appropriate.

Thanks for letting me review this manuscript.

This could be a nice paper.

Level of interest: An article whose findings are important to those with closely related research interests.

Quality of written English: Well.

Statistical review: No.

Declaration of competing interests:

I declare that I have no competing interest.

Reviewer #2: 

The answer of "no" to Question 4 is only referring to a few minor grammatical issues flagged in the attachment. This is a well-written paper overall, but according to the review instructions I would choose "yes" only if I had no revisions at all.

The answer of "no" to Question 3: The instructions state that the data should be included as a supplemental file or publicly available. I was not able to find this in the supplemental file that was included with this submission, nor could I find a website that makes the data available - my apologies if I missed it. I do see your statement that the data will be made available - I will leave it to the editor if this can simply be added as a revision to this paper.

R. In this current version of the manuscript the dataset was uploaded as Supporting Information (S1. Dataset).

Overall, the aim of this paper was to investigate whether hearing aid self-efficacy is related to educational level, selecting a population from a Latin American country on the argument that this population may vary in educational level (which turned out to be true in this sample). The self-efficacy measure is published, well-constructed, and has been applied successfully in several past hearing aid studies on a range of topics. The authors applied a rigorous, well-described translation process to this questionnaire in order to develop a Spanish-language version aiming to maintain the questionnaire’s items and scoring methods. The authors also piloted this translation in a sample of 48 participants to ensure that it was feasible for use, prior to applying it in the current study in a well-justified larger sample. In this sample, the authors performed data collection and analysis to answer the primary aim of the study, relating hearing aid self-efficacy to educational level, among several other factors. Overall, this method is reasonable and the question is of interest in our understanding of factors that relate to hearing aid self-efficacy, which on its own has been shown to impact rehabilitation outcomes. It is not surprising that the authors found a relationship between hearing aid self-efficacy and educational level.

This is a well-conducted study that makes a good contribution to the literature, and may provide guidance to those who provide hearing rehabilitation services or those who develop hearing aid user’s guides for a wide range of populations.

My only concern with this paper is its actual breadth versus its title. With a full translation, feasibility evaluation, and confirmatory factor analysis to not only develop but also to validate the Spanish version of the MARS-HA, this paper has not one but two primary aims. Only the latter is identified in the title. 

R. The title was modified with the aim of including the validation process of the Spanish version of the MARS-HA questionnaire:

Does Educational Level Predict Hearing Aid Self-Efficacy in Experienced Older Adult Hearing Aid Users from Latin America? Validation process of the Spanish version of the MARS-HA questionnaire.

Minor comments:

Page 6, 2nd paragraph: 

I don’t think the translated version can be considered “preliminary” and “final” at the same time.

R. The word “final” was not really what we meant. We just wanted to mention “preliminary version”. Therefore, the word “final” was deleted from this statement (see page 7, line 12).

“Equivalencies” or “equivalence” perhaps rather than “equivalences”

R. The word “equivalences” was replaced by “equivalence” (see page 7, line 13).

Page 9, bottom:

“hours” rather than “hour”

R. The word “hour” was replaced by “hours” (see page 10, line 17). 

Page 11, adherence to hearing aids

You have chosen to use a self-report measure of adherence rather than the data log of hearing aid use provided by the hearing aid. Please justify this choice (perhaps reading data logs was not feasible in the context of home visits) and include literature (I know of one 2014 study by Laplante-Leveque et al) on the validity of self-report. Given the large reporting ranges used on the IOI, minor differences between data logs and self report may not have impacted your analyses, but it would be helpful to the reader to acknowledge any limitations of self-report use metrics, either here or in the Discussion.

R. Hearing aid data-logging could not be accessed in this study as data collection was carried out at participants’ homes. We have now provided this explanation in the revised version of the manuscript, including a reference about how accurate self-report metrics can be in this field of research (see page 36, lines 7-14)

References

- Laplante-Lévesque A, Nielsen C, Jensen LD, Naylor G. Patterns of hearing aid usage predict hearing aid use amount (data logged and self-reported) and overreport. J Am Acad Audiol. 2014; 25(2): 187-98.

Page 27

The sentence starting with “The fact that these pairs of items” needs some editing. I think “other than the factor with which they loaded” might be what you meant.

R. The sentence “other than the factor they loaded” was replaced by “other than the factor with which they loaded” (see page 28, lines 9-10).

Participants were fitted with hearing aids that did not offer noise management. Why was this unusual fitting profile selected? Is this due to the aids that were available (unlikely) or was it a decision to not provide this? Do you think this explains your low scores on aided listening?

R. In Chile, adults aged 65 and older who require hearing aids are provided with one device (if they belong to the public healthcare system) either for free or with a maximum co-payment of 20% of the device price (Government of Chile). The latter is determined based on the person’s income levels. Under this programme, hearing aids with multiple channels and up to four programmes whose cost is low (approximately $105 USD) are provided by public hospitals. Each hospital puts out a tender for the contract to purchase the hearing aids and the company chosen fits them and perform follow-up appointments with the patient (this involves three sessions to adjust the hearing aid and teach basic aspects of its care and use). In most cases, the dispenser did not activate the noise management programmes nor adapted a hearing aid with adaptive noise cancellation algorithms.

We believe that the aided listening subscale had relatively lower scores because few older adults participated in social activities. Thus, participating in more social activities allows people to experience a richer variety of listening environments providing a sort of listening training with the hearing aid. So the more social activities the older adults participate in while wearing the hearing aid, the better their performance in daily life listening activities, which should be reflected by better scores for the aided listening S-MARS-HA subscale (see page 33, lines 1-11).

References

- Government of Chile [Gobierno de Chile]. Superintendencia de Salud. Problema de salud AUGE N° 56. Hipoacusia Bilateral en personas de 65 años y más que requieren uso de audífono. [Last accessed on 2019 October 16]. Available from: http://www.supersalud.gob.cl/difusion/572/w3-article-3710.html

Page 28

The sentence ending with “a better hearing aid self-efficacy” needs some editing. Perhaps “… older adults with higher educational levels exhibited higher hearing aid self-efficacy”. 

R. The sentence “older adults with higher educational levels could reach a better hearing aid self-efficacy” was replaced by “older adults with higher educational levels exhibited higher hearing aid self-efficacy” (see page 29, line 23, and page 30 line 1).

Page 28, last paragraph

When I first read “readability of the written materials” I initially thought you were referring to the S-MARS-HA but as I read on, I realized you mean the hearing aid instruction manuals. Please clarify, perhaps “readability of the written materials (e.g., hearing aid user’s guide)”.

R. The sentence was modified according to the reviewer’s comment (see page 30, lines 4-5).

My co-authors and I would like to thank you and your review team for their careful reading of our manuscript and the valuable comments offered. We feel they have led to an improved version and we look forward to receiving your further advice regarding the status of our manuscript.

---

## [Decision Letter · Decision Letter 1]

20 Nov 2019

Does Educational Level Predict Hearing Aid Self-Efficacy in Experienced Older Adult Hearing Aid Users from Latin America? Validation process of the Spanish version of the MARS-HA questionnaire

PONE-D-19-19089R1

Dear Dr. Fuentes-López,

We are pleased to inform you that your manuscript has been judged scientifically suitable for publication and will be formally accepted for publication once it complies with all outstanding technical requirements.

With kind regards,

Stefano Federici, Ph.D.

Academic Editor

PLOS ONE

Additional Editor Comments (optional):

Reviewers' comments:

Reviewer's Responses to Questions

**Comments to the Author**

1. If the authors have adequately addressed your comments raised in a previous round of review and you feel that this manuscript is now acceptable for publication, you may indicate that here to bypass the “Comments to the Author” section, enter your conflict of interest statement in the “Confidential to Editor” section, and submit your "Accept" recommendation.

Reviewer #1: All comments have been addressed

Reviewer #2: All comments have been addressed

2. Is the manuscript technically sound, and do the data support the conclusions?

Reviewer #1: Yes

Reviewer #2: Yes

3. Has the statistical analysis been performed appropriately and rigorously? 

Reviewer #1: Yes

Reviewer #2: Yes

4. Have the authors made all data underlying the findings in their manuscript fully available?

Reviewer #1: Yes

Reviewer #2: (No Response)

5. Is the manuscript presented in an intelligible fashion and written in standard English?

Reviewer #1: Yes

Reviewer #2: Yes

6. Review Comments to the Author

Reviewer #1: This is a cross sectional observational study in a convenience sample from an urban hearing impairment outpatient program of Governmental hospital in Santiago (Chile). This paper has three components, one is the validity of a questionnaire used to measure hearing aid self-efficacy: the Spanish version of Measure of Audiologic Rehabilitation Self-Efficacy for Hearing Aids (S- MARS-HA); another second component about the role of educational level and other factors as attitudes in using hearing aids and the pure-tone average, (PTA) in the fitted ear as important factor of hearing aid self efficacy and, finally the third component of this manuscript explores the association between hearing aid self-efficacy with adherence to hearing aid use. The topic of the manuscript is appropriate for the Journal. It could be of interest to investigators and clinicians.

All comments and doubts have been clarified. All recomendantions were included in the main text. So, the paper could be accepted for publishing.

Reviewer #2: (No Response)

7. PLOS authors have the option to publish the peer review history of their article (what does this mean?). If published, this will include your full peer review and any attached files.

Reviewer #1: No

Reviewer #2: No

---

## [Editor Report · Acceptance letter]

11 Dec 2019

PONE-D-19-19089R1 

Does Educational Level Predict Hearing Aid Self-Efficacy in Experienced Older Adult Hearing Aid Users from Latin America? Validation process of the Spanish version of the MARS-HA questionnaire 

Dear Dr. Fuentes-López:

I am pleased to inform you that your manuscript has been deemed suitable for publication in PLOS ONE. Congratulations! Your manuscript is now with our production department. 

With kind regards,

on behalf of

Prof. Stefano Federici 

Academic Editor

PLOS ONE